# A neural network approach to predict opioid misuse among previously hospitalized patients using electronic health records

**Lucas Vega** [1]\*, **Winslow Conneen**[1☯], **Michael A. Veronin**[2☯], **Robert P. Schumaker**[3☯]

**1** Data Analytics Lab, The University of Texas at Tyler, Tyler, Texas, United States of America,
**2** Pharmaceutical Sciences Department, The University of Texas at Tyler, Tyler, Texas, United States of America, **3** Computer Science Department, The University of Texas at Tyler, Tyler, Texas, United States of America

☯ These authors contributed equally to this work.
\* Lucasvega771@gmail.com

**Data Availability Statement:** Data cannot be shared publicly because [the data underlying the results of our study (MIMIC-IV Version 1.0) contains individual patient records and is thus a

## Abstract

Can Electronic Health Records (EHR) predict opioid misuse in general patient populations? This research trained three backpropagation neural networks to explore EHR predictors using existing patient data. Model 1 used patient diagnosis codes and was 75.5% accurate. Model 2 used patient prescriptions and was 64.9% accurate. Model 3 used both patient diagnosis codes and patient prescriptions and was 74.5% accurate. This suggests patient diagnosis codes are best able to predict opioid misuse. Opioid misusers have higher rates of drug abuse/mental health disorders than the general population, which could explain the performance of diagnosis predictors. In additional testing, Model 1 misclassified only 1.9% of negative cases (non-abusers), demonstrating a low type II error rate. This suggests further clinical implementation is viable. We hope to motivate future research to explore additional methods for universal opioid misuse screening.

## Introduction

In the United States, nearly 10 million individuals misuse opioids annually [1]. Opioid misuse occurs when an individual 1) takes opioids in different/larger quantities than prescribed by their physician, or 2) takes opioids without a prescription [2]. Prolonged opioid misuse often leads to an opioid use disorder (OUD) which describes the problematic pattern of opioid use that causes significant impairment or distress. Indicators of OUD within a patient include, but are not limited to, strong cravings for opioids, increased opioid tolerance, withdrawal symptoms when regular opioid consumption stops and difficulties performing tasks at work/school [3]. Widespread OUD has resulted in an ongoing epidemic, with over 75,000 opioid overdose deaths between April 2020-April 2021 [4]. The opioid epidemic began in the early 2000s when medical practices sharply increased the number of opioid-based prescriptions as an effective means to treat chronic pain [5]. Since then, the misuse of opioids has contributed to a significant portion of reported adverse drug events [6]. In 2016, the CDC released an opioid

restricted-access resource]. Data are available from the PhysioNet Institutional Data Access / Ethics Committee (contact via email: contact@physionet.org or follow this link https://physionet.org/content/mimiciv/view-required-training/1.0/#1 to complete the required training) for researchers who meet the criteria for access to confidential data. The data underlying the results presented in the study are available from (PhysioNet and can be obtained through the following link: https://physionet.org/content/mimiciv/1.0/ This link references a web page that clearly details all the necessary steps to access the data).

**Funding:** The author(s) received no specific funding for this work.

**Competing interests:** The authors have declared that no competing interests exist.

prescribing guideline, which encouraged medical professionals check to patient health records for opioid misuse risk factors before prescribing [7]. However, this alone is insufficient, as patients commonly have conditions outside of these datasets. Therefore, before prescribing opioids, medical professionals often administer questionnaires/surveys to screen patients for undocumented risk factors [8]. Previous research has developed a variety of questionnaires/surveys towards this end. The list includes, but is not limited to, the Opioid Risk Tool (ORT) [9], the Screener and Opioid Assessment for Patients with Pain-Revised (SOAPP-R) [10], the Brief Risk Interview/Questionnaire [11] and the Prescription Opioid Therapy Questionnaire [12]. Of these, the ORT and SOAPP-R are generally recommended over other tools [13]. The implementation of these tools helped adjust prescribing practices. As a result, opioid prescriptions have decreased by 44% over the last decade [14].

Despite this change, opioid overdose rates have continued to increase [14]. This is partially due to individuals misusing opioids outside of a prescription. Additionally, the use of illicit opioids has increased, as heroin is involved in roughly 15,000 annual overdoses [1]. These facts make identifying and treating opioid misuse challenging for medical practices. Any patient could potentially be an opioid misuser, but not all patients present visible signs of misuse until an overdose occurs. Thus, the thorough screening of all patients (universal screening) is necessary to address this problem. Questionnaires cannot support universal screening, as their administration and review process is too time consuming. Therefore opioid risk assessment is currently limited to those being considered for prescriptions. This motivates the development of faster screening methods. One potential approach involves machine learning (ML), or computer algorithms designed to learn patterns from datasets in order to produce a model. Specifically, these algorithms approximate a relationship between different feature sets and can be used to make predictions. If a ML model could be trained to predict opioid misuse with available medical records, it could automatically screen all patients within the system. Patients identified as "at-risk" by the model could then be further evaluated by a medical professional. Previous research has explored developing ML models to predict/identify various opioid use outcomes. These outcomes include prolonged opioid use ($\geq$ 90 days since first prescription) [15–17], an opioid overdose [18, 19], opioid misuse [20] and opioid dependence (presence of withdrawal symptoms in absence/reduced amounts of opioid use) [21–24]. These studies have reported positive results, which suggests that reliable prediction of opioid use outcomes is possible. However, the majority of these studies used national survey responses or insurance claims to train their ML model(s). Most institutions lack access to these datasets, which presents a barrier to implementation. Developing a ML model using a medical practices' own Electronic Health Records (EHR) could overcome this problem. An EHR dataset is a digital representation of a patient's paper chart [25]. These datasets contain patient diagnosis history, prescription history, procedure history, admission times, and other healthcare related features.

In our review of existing opioid use outcome models trained with EHR data, we made note of the following research gaps. First, no model was designed to predict opioid misuse in general patient populations, where universal screening would occur. With the exception of work done by Ellis et al., (which focused primarily on opioid dependence) [22] previous research considered only prescription opioid users. This excludes patients without prescriptions that misuse/don't misuse opioids. Second, diagnoses and prescriptions have not been explored as OUD predictors individually. Previous research has demonstrated that these features in combination can reliably predict OUD conditions [22]. To expand the knowledge of EHR predictors, models should be constructed with exclusively diagnosis features or exclusively prescription features, and then compared to models utilizing these features in combination. Finally, in the related study considering general patient populations (Ellis et al.) only a Random forests model was trained. This motivates exploring the utility of other models. A

backpropogation neural network (BPNN) is a machine learning model that organizes a network of artificial neurons to learn from data in a manner that mimics the human brain [26]. In recent years, BPNN models have demonstrated superior performance over traditional ML methods in a variety of tasks. These include, but are not limited to, landslide prediction, [27] modeling quantum circuits, [28] computer vision [29] and predicting healthcare outcomes via EHR datasets [30].

From these gaps we seek to extend our knowledge by asking the following research questions.

*1. Given a patient's EHR diagnosis history, can a neural network model predict opioid misuse in general patient populations?*

To explore prediction in larger populations, we seek to evaluate the performance of ML models utilizing diagnosis features. We anticipate that reliable prediction is possible, as opioid misusers are likely to have additional mental health/drug abuse diagnoses that differentiate them from non-misusers.

*2. Given a patient's EHR prescription history, can a neural network model predict opioid misuse in general patient populations?*

To explore prediction in larger populations, we seek to evaluate the performance of ML models utilizing prescription features. We previously hypothesized that diagnosis history could enable reliable prediction. As diagnoses are often accompanied by drug treatment, we also anticipate reliable prediction with prescription history.

*3. Will a neural network model using both EHR diagnosis and prescription data perform better in predicting opioid misuse in general patient populations than models using either diagnoses or prescriptions exclusively?*

We are curious to observe the change in predictive power that results from utilizing both diagnoses and prescriptions as features, as opposed to individually. We anticipate models using these features in combination will perform better than models using only diagnoses or prescriptions.

## Materials and methods

To answer our research questions, we constructed the Opioid Misuse Prediction System (OMPS) as shown in Fig 1. In this section, we detail the major features of OMPS and report findings specific to our project. The first subsection presents the dataset used in our experiment. The remaining subsections detail the Misuser Cohort Selection, Identification of Common Features, Non-Misuser Cohort Selection, LASSO Feature Selection, Final Cohort Selection, BPNN and the Evaluation Metrics.

### Data

MIMIC-IV (Medical Information Mart for Intensive Care, version 4) is an EHR dataset of de-identified patient records from Beth Israel Deaconess Medical Center in Boston, Massachusetts [31]. The dataset is freely available for research (mimic.mit.edu) and contains the patient health records data for 382,278 individuals between 2008-2019.

The data structure of MIMIC-IV is organized into six main modules: hosp, core, icu, ed, cxr, and note. The *hosp* module refers to hospital patient data such as diagnoses, labs and medications. The *core* module contains patient tracking information, including admission/transfer dates and demographic data. The *icu* module refers to the intensive care unit, including charts. The *ed* module refers to the emergency department such as admission, triage, vitals and medications. The *cxr* module refers to radiology reports. Finally, the *note* module contains free-text clinical notes, but is currently not publicly available.

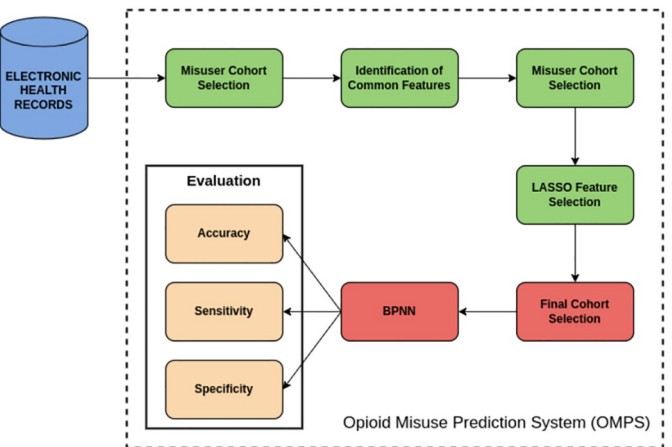

**Fig 1. Opioid Misuse Prediction System.**

For our experiment, we used the *hosp* and *core* modules. From *hosp* we used the prescriptions, pharmacy and diagnoses_icd tables. Prescriptions provided data on the known patient prescriptions, whereas diagnoses_icd provided diagnosis codes. The International Classification of Diseases—Clinical Modification (ICD-CM) was used to encode the diagnoses. ICD-CM is a classification standard used to code and classify patient morbidity data [32]. The pharmacy table provided additional information concerning drugs prescribed to a patient. From *core* we used the admissions table, which provides the dates associated with individual patient hospital visits. The tables in *hosp* and *core* were used to identify opioid misusers and non-misusers, and subsequently gather and chronologically order their diagnosis and prescription history. Patients were uniquely identified by the anonymized subject_id field.

For our experiment, we used ICD-CM version 9 diagnosis codes. MIMIC-IV contains multiple diagnosis coding schemes, which can introduce accuracy and consistency issues [33]. From this, we chose to only consider one version. We chose ICD-CM-9 over ICD-CM-10 as it was more abundant in the dataset.

## Misuser cohort selection

To identify opioid misusers in the dataset, we gathered the subject_id of all patients who had an ICD-CM-9 code for non-dependent opioid misuse (305.5*) in the diagnoses_icd table. Since our aim is to identify opioid misusers before their condition worsens, non-dependent opioid misuse was considered to be an ideal code to focus on, as we considered it to be the earliest and least severe marker of OUD. There are four distinct ICD-CM-9 codes for non-dependent opioid misuse, as some codes provide additional detail concerning the diagnosis. There is opioid abuse-unspecified (305.50), opioid abuse-continuous (305.51), opioid abuse-episodic (305.52), and opioid abuse-in remission (305.53). Our opioid misuser cohort included patients with at least one of these codes in their medical history. From these four codes, we identified 1,619 opioid misusers.

Following this, the diagnosis and prescription history of each opioid misuser was recorded. We restricted patient medical history to only include data prior to the hospital visit where the patient was first diagnosed with opioid misuse. If an opioid misuser had no diagnosis history or no prescription history after applying this restriction, they were excluded from the cohort. This restriction ensures each of our predictive models can be trained and tested with the same

cohort, as some models use prescription/diagnosis data exclusively. Ensuring our models use the same cohort allows us to compare different feature sets. Applying these restrictions left us with 566 opioid misusers in our cohort.

## Identification of common features

Our next step was to identify the diagnoses/prescriptions that could serve as strong features in our predictive models. We started by investigating the diagnoses and prescriptions that were the most commonly found within the medical history of opioid misusers. The diagnoses and prescriptions were stored in separate tables, and subsequently ordered by the number of opioid misusers who had the given diagnosis/prescription in their history. OUD diagnoses and opioid prescriptions were removed from these tables. We removed these features in anticipation of training our predictive models. We wanted our models to learn which non-opioid/non-OUD features are indicative of misuse, to better identify at-risk patients with no documented history of OUD. We then shortened each table to include only the top 50 most common features.

Before their removal, 304.01 (opioid dependence continuous) and 304.00 (opioid dependence unspecified) were the 9th and 14th most common opioid misuser diagnoses, respectively. Similarly, before their removal, the opioid prescriptions Oxycodone (immediate release), Hydromorphone (Dilaudid), Morphine Sulfate, Oxycodone-Acetaminophen, and Tramodol (Ultram) were the 9th, 11th, 18th, 40th, and 46th most common opioid misuser prescriptions respectively. Despite being an opioid, Methadone was not removed from the list because the medication is not prescribed to treat chronic pain. Rather, it used to treat OUD.

## Non-misuser cohort selection

Three non-misuser cohorts were gathered in preparation for three LASSO tests. The rationale and structure of the LASSO tests will be discussed in greater detail in later sections. To identify non-misusers in the dataset, we gathered the subject_id of all patients who did not have any ICD-CM-9 codes associated with OUD in the diagnoses_icd table. These codes are listed in Table 1.

**Table 1. Opioid use disorder ICD-9 codes.**

| ICD-9 | ICD-9 Description |
|-------|-------------------|
| 304.00 | opioid dependence-unspecified |
| 304.01 | opioid dependence continuous |
| 304.02 | opioid dependence-episodic |
| 304.03 | opioid dependence in remission |
| 304.70 | combinations of opioid type drug with any other drug dependence-unspecified |
| 304.71 | combinations of opioid type drug with any other drug dependence-continuous |
| 304.72 | combination of opioid type drug with any other drug dependence-episodic |
| 304.73 | combinations of opioid type drug with any other drug dependence-in remission |
| 305.50 | opioid abuse-unspecified |
| 305.51 | opioid abuse-continuous |
| 305.52 | opioid abuse-episodic |
| 305.53 | opioid abuse-in remission |
| 965.00 | opium poisoning |
| E85.02 | accidental poisoning by other opiates and related narcotics |
| E93.52 | other opiates and related narcotics causing adverse effects in therapeutic use |

From this, we identified 380,659 non-misusers. Then, three cohorts of 566 randomly selected non-misusers were created. The first cohort contained non-misusers with at least one entry from the top 50 most common opioid misuser diagnoses table in their medical history. We referred to this group as the diagnosis non-misuser cohort. The second cohort had a similar inclusion rule, but for the top 50 most common opioid misuser prescriptions table. We referred to this group as the prescription non-misuser cohort. The diagnosis history of the first cohort and the prescription history of the second cohort were recorded. Data from the first non-misuser cohort were used in LASSO test to identify strong predictors of opioid misuse among diagnoses. Data from the second cohort was used in a similar test, but for prescriptions. The results of these tests were used in the construction of the third non-misuser cohort; the combined non-misuser cohort. The next section explains this process in greater detail.

## LASSO feature selection

Three least absolute shrinkage and selection operator (LASSO) logistic regression tests were run for feature selection. We chose LASSO due to its resistance to overfitting and ability to reduce dimensionality among sets of highly correlated variables [34]. Opioid misuse was the target variable for each test. Test 1 analyzed diagnosis data from the opioid misuser and diagnosis non-misuser cohorts. The independent variable set was the top 50 most common opioid misuser diagnoses. Test 2 analyzed prescription data from the opioid misuser and prescription non-misuser cohorts. Here, the independent variable set was the top 50 most common opioid misuser prescriptions.

Test data was represented as binary matrices. Each row represented an individual patient's medical features. The last column identified the patient as an opioid misuser/non-misuser (1 = opioid misuser, 0 = non-misuser). The remaining columns indicated the existence of a diagnosis/prescription within a patient's medical history (1 = exists, 0 = does not exist).

From each test, the top 10 diagnoses/prescriptions with the highest odds-ratio were recorded. Table 2 contains the recorded results from Test 1. Table 3 contains the recorded results from Test 2.

In preparation for a third LASSO test (Test 3), we gathered the combined non-misuser cohort. Here, non-misusers were required to have at least one entry from both of the above tables. Test 3 analyzed diagnosis and prescription data from the opioid misuser cohort and combined non-misuser cohort. The independent variable set was the 20 diagnoses/prescriptions identified from the previous LASSO tests. We again recorded the 10 features with the highest odds-ratios, which are displayed in Table 4.

**Table 2. Top 10 most significant diagnoses.**

| Num | ICD-9 | ICD-9 Description | Odds-Ratio |
|---|---|---|---|
| 1 | 292.0 | Drug Withdrawal | 3.12 |
| 2 | 070.54 | Chronic hepatitis C without mention of hepatic coma | 2.47 |
| 3 | 070.70 | Unspecified viral hepatitis C without hepatic coma | 2.09 |
| 4 | 309.81 | Posttraumatic stress disorder | 1.42 |
| 5 | 305.1 | Tobacco Use Disorder | 1.31 |
| 6 | 300.4 | Dysthymic Disorder | 1.27 |
| 7 | 305.90 | Unspecified Drug Abuse | 1.21 |
| 8 | 305.60 | Cocaine abuse, unspecified | 1.10 |
| 9 | 338.29 | Other chronic pain | 1.05 |
| 10 | 291.81 | Alcohol withdrawal | 0.93 |

**Table 3. Top 10 most significant prescriptions.**

| Num | Prescription | Odds-Ratio |
|---|---|---|
| 1 | Methadone | 2.59 |
| 2 | Nicotine Patch | 1.80 |
| 3 | Diazepam | 1.46 |
| 4 | Zolpidem Tartrate | 1.16 |
| 5 | Quetiapine Fumarate | 1.12 |
| 6 | Clonazepam | 1.06 |
| 7 | 0.45% Sodium Chloride injection | 0.69 |
| 8 | Lorazepam | 0.67 |
| 9 | Thiamine | 0.63 |
| 10 | Gabapentin | 0.47 |

## Final cohort selection

The recorded diagnoses/prescriptions from each LASSO test were chosen to be the features of our three predictive models. The next step was to select our cohorts for training/testing. We removed patients from the opioid misuser cohort who did not have an entry from both Tables 2 and 3 in their medical history. This left us with 465 opioid misusers. A final cohort of 465 non-misusers was also gathered. The inclusion rule for the non-misuser cohort was the same as the opioid misuser cohort. The demographics of both cohorts is displayed in Table 5. We report sex, age, race, ethnicity, marital status, insurance status and whether or not the patient was prescibed opioids.

We excluded opioid misusers/non-misusers with none of our model's features to avoid training the model to associate an empty input set with opioid misuse/non-misuse. Our model needs to learn the specific combination of inputs that are associated with opioid misuse to be effective at distinguishing opioid misusers from non-misusers. Patients with none of our inputs could not be evaluated with this model.

## BPNN

We trained three separate ten-fold backpropagation neural networks to predict opioid misuse. All models utilized a logistic sigmoid activation function. All models used the final opioid misuser/non-misuser cohorts. Training and testing data were represented as three binary matrices. Each matrix corresponded to a different model. The structure of the matrices was similar

**Table 4. Top 10 most significant combined features.**

| Num | Diagnosis/Prescription | Odds-Ratio |
|---|---|---|
| 1 | (292.0) Drug Withdrawal | 2.79 |
| 2 | Methadone | 1.63 |
| 3 | (070.70) Unspecified viral hepatitis C without hepatic coma | 1.55 |
| 4 | (070.54) Chronic hepatitis C without mention of hepatic coma | 1.42 |
| 5 | (305.90) Unspecified Drug Abuse | 1.07 |
| 6 | (305.60) Cocaine abuse, unspecified | 0.82 |
| 7 | Nicotine Patch | 0.82 |
| 8 | (309.81) Posttraumatic stress disorder | 0.73 |
| 9 | (338.29) Other Chronic Pain | 0.46 |
| 10 | (291.81) Alcohol withdrawal | 0.38 |

**Table 5. Machine learning cohort demographics.**

| Demograpic | Misusers | Non-Misuser |
|---|---|---|
| **Sex** | | |
| Male | 62% | 53% |
| Female | 38% | 47% |
| **Age** | | |
| 18-24 | 21% | 16% |
| 35-54 | 59% | 38% |
| 55-74 | 19% | 35% |
| >74 | <1% | <1% |
| **Race** | | |
| White | 75% | 76% |
| Black/African American | 20% | 12% |
| Asian | <1% | <1% |
| American Indian/Alaska Native | <1% | <1% |
| Other | 4% | 5% |
| Unknown/Unable to Obtain | <1% | 4% |
| **Ethnicity** | | |
| Other | 91% | 95% |
| Hispanic/Latino | 8% | 5% |
| Unknown/Unable to Obtain | <1% | <1% |
| **Marital Status** | | |
| Single | 69% | 58% |
| Married | 15% | 31% |
| Divorced | 11% | 11% |
| Widowed | 4% | 5% |
| Unknown/Unable to Obtain | <1% | 0% |
| **Insurance** | | |
| Medicare/Medicaid | 72% | 51% |
| Other | 28% | 49% |
| **Prescribed Opioids** | | |
| Yes | 94% | 92% |
| No | 6% | 8% |

to the matrices for the LASSO tests. Each row represents a patient. The last column identifies the patient as an opioid misuser/non-misuser. The remaining columns of each matrix contain the feature set of the corresponding model.

**Model 1—Diagnosis history.** We trained/tested Model 1 to answer research question 1: *Given a patient's EHR diagnosis histroy, can a neural network model predict opioid misuse in general patient populations?*

This model was constructed to predict opioid misuse using patient diagnosis history. The feature set for Model 1 was entries from the 10 significant diagnoses table (Table 2). The training/testing matrix represented the diagnosis history of our final opioid misuser and non-misuser cohorts.

**Model 2—Prescription history.** We trained/tested Model 2 to answer research question 2: *Given a patient's EHR prescription histroy, can a neural network model predict opioid misuse in general patient populations?*

This model was constructed to predict opioid misuse using patient prescription history. The feature set for Model 2 was entries from the 10 significant prescriptions table (Table 3). The training/testing matrix represented the prescription history of our final opioid misuser and non-misuser cohorts.

**Model 3—Diagnosis and prescription history.** We trained/tested Model 3 to answer research question 3: *Will a neural network model using both EHR diagnosis and prescription data perform better in predicting opioid misuse in general patient populations than models using either diagnoses or prescriptions exclusively?*

This model was constructed to predict opioid misuse using patient diagnosis and prescription history. The feature set for Model 3 was entries from the 10 significant diagnoses and prescriptions table (Table 4). The training/testing matrix represented the prescription history of our final opioid misuser and non-misuser cohorts.

**Evaluation metrics.** The performance of each model was evaluated using ten-fold cross validation. From these tests, we recorded accuracy, sensitivity and specificity scores. These scores were used to evaluate and compare the predictive power of each model's feature set.

## Results and discussion

### Research question 1

*Given a patient's EHR diagnosis histroy, can a neural network model predict opioid misuse in general patient populations?*

Our results suggest diagnosis codes alone can reliably predict opioid misuse. Model 1 demonstrated good accuracy (75.5%), fair sensitivity (70.7%), and good specificity (80.2%). The performance of the diagnosis feature set could be explained by its containment of OUD risk factors. Seven items indicated either substance abuse (Drug withdrawal, Unspecified drug abuse, Tobacco Use Disorder, Cocaine abuse, and Alcohol withdrawal) or mental illness (Post-traumatic stress disorder; Dysthymic Disorder) within a patient. Additionally, two separate hepatitis C diagnoses (Chronic hepatitis C without mention of hepatic coma; Unspecified viral hepatitis C without hepatic coma) were present, which is a common infection among drug misusers. Chronic pain was identified as a prescription significant for distinguishing misusers from non-misusers. We postulate this is because chronic pain is associated with long-term opioid use. The longer one uses opioids, the more likely they are to develop OUD and misuse the drug.

Model 1 utilized no OUD diagnoses, as they were excluded from feature engineering. Patients with recorded OUD diagnoses could be identified with EHR querying. However, opioid misusers without these features may go undetected. This fact motivated us to explore other predictors. Despite this, 9 of 10 entries in our feature set are well-known OUD risk factors. Hence, Model 1 may provide little utility in identifying at-risk patients during the prescription process, where doctors ideally check records. For screening in large populations, where manual EHR querying is impractical, the performance of Model 1 is encouraging. Minimizing the type II error rate is critical for clinical implementation. Frequent misclassification of non-misusers would waste hospital resources. The strong specificity score (80.2%) indicates a low false positive rate, suggesting resource efficient screening is viable.

Our training cohorts were both required to have at least one entry from the diagnosis feature set in their medical history. From this, our models likely learned to identify opioid misusers among at-risk populations. Fig 2 illustrates the distribution of diagnoses in our training cohorts. Opioid misusers and non-misusers had similar rates of Tobacco Use Disorder and Dysthymic Disorder. Hepatitis C and other drug abuse diagnoses (Drug withdrawal, Unspecified drug abuse, Cocaine abuse unspecified) were significantly more common in the opioid

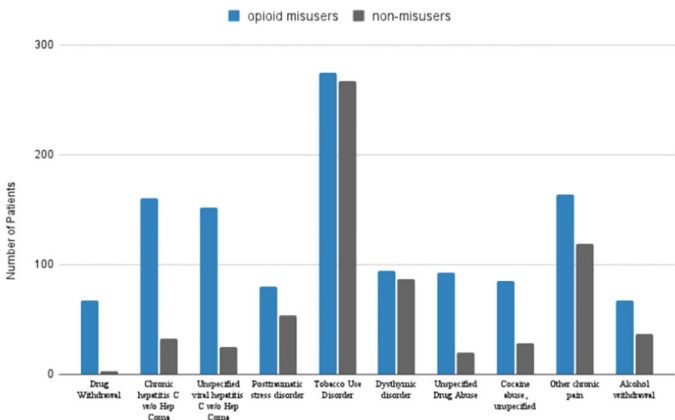

**Fig 2. Distribution of diagosis features in training cohort.**

misuser cohort. Their presence suggests that these features may distinguish opioid misusers from other at-risk populations.

## Research question 2

*Given a patient's EHR prescription histroy, can a neural network model predict opioid misuse in general patient populations?*

Model 2 demonstrated the worst accuracy (64.9%) of the three models. Specificity was fair (72.0%), but offset by poor sensitivity (57.7%). This indicates Model 2 frequently misidentified opioid misusers as non-misusers. Fig 3 illustrates the distribution of prescriptions in our training cohorts.

After observing Figs 2 and 3, we postulated opioid misusers and non-misusers differed more by their diagnosis history than prescription history. We ran two Chi-Squared independence tests to quantify these differences. Test 1 examined the feature distribution of diagnoses. Test 2 was similar but for prescriptions. P-values from Tests 1 and 2 were 2.2e-16 and 1.03e-10 respectively. Both tests indicate significant differences in feature distributions between the two cohorts.

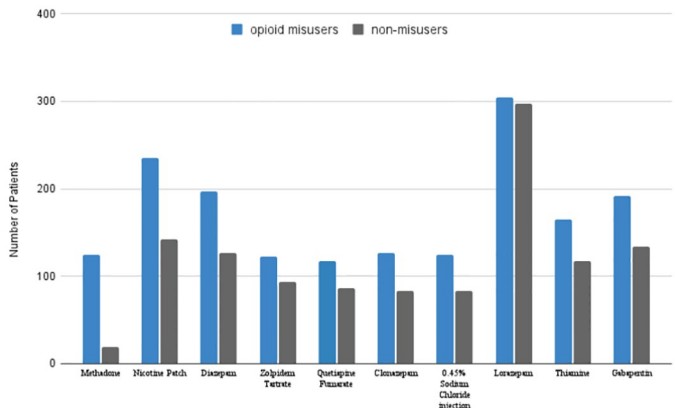

**Fig 3. Distribution of prescription features in training cohort.**

It's worth highlighting that our non-misuser cohort contained many non-opioid drug misusers. Model 2 could potentially perform better in more general populations, where drug abuse is less common. Our prescription feature set contained six medications to treat various mental health disorders. Zolpidem Tartrate is utilized in short term treatment of insomnia. Quetiapine Fumarate is an antipsychotic used for the treatment of schizophrenia, bipolar disorder and major depressive disorder. Clonazepam and Gabapentin are both anticonvulsants used to treat seizures and panic attacks. Finally, Lorazepam is used to treat anxiety. As mental health issues are increasing worldwide, the administration of these medications is becoming more common. From this, our feature set may struggle to distinguish opioid misusers from non-misuser populations.

### Research question 3

*Will a neural network model using both EHR diagnosis and prescription data perform better in predicting opioid misuse in general patient populations than models using either diagnoses or prescriptions exclusively?*

Model 3 demonstrated better accuracy (74.5%), sensitivity (72.6%), and specificity (76.3%) than Model 2. This suggests diagnosis and prescription history in combination performs better than prescription history alone. However, Models 1 and 3 performed similarly. Model 1 had slightly higher accuracy (75.5% vs 74.5%) and higher specificity (80.2% vs 76.3%), while Model 3 had slightly higher sensitivity (72.6% vs 70.7%). From this, our results do not suggest combining diagnoses and prescriptions will outperform diagnoses alone. As stated earlier, strong specificity is important for clinical implementation. From this, Model 1 may be preferable to Model 3. Additionally, many hospitals have limited data storage/processing capacity. Requiring only diagnosis codes reduces computation cost, making model development more viable in these settings.

The feature sets of Models 1 and 3 differed by only two items. Model 1 utilized Tobacco Use Disorder and Dysthymic Disorder, while Model 3 instead utilized Methadone and Nicotine Patch. We initially believed this change would cause Model 3 to outperform Model 1. A Nicotine Patch suggests drug addition, while a Methadone prescription is indicative of OUD. Moreover, Fig 2 clearly illustrates that Tobacco Use Disorder and Dysthymic Disorder appear in similar rates between opioid misusers and non-misusers, suggesting these features provide little predictive utility. Model 1 outperformed Model 3 in specificity (80.2% vs 76.3%) and overall accuracy (75.5% vs 74.5%). This suggests diagnosis combinations involving Tobacco Use Disorder or Dysthymic Disorder were important for classifying negative cases. For example, Dysthymic Disorder, Nicotine Patch, and Alcohol Withdrawal may suggest opioid misuse, while Tobacco Use Disorder, Nicotine Patch and Alcohol Withdrawal may suggest non-opioid drug misuse. Model 3 only outperformed Model 1 in sensitivity (72.6% vs 70.7%). These results suggest swapping diagnosis codes for prescriptions decreases false negatives, but increases false positives. Perhaps combining these features, instead of swapping, would increase both sensitivity and specificity. To test this, we developed a new BPNN; Model 4. The feature set of Model 4 was the same as Model 3, but additionally included Tobacco Use Disorder and Dysthymic Disorder. The machine learning parameters and training/testing cohorts of Model 4 were the same as the other models. In ten-fold cross validation, Model 4 demonstrated 75.0% accuracy, 71.2% sensitivity and 78.9% specificity. Compared to swapping (Model 3), combining features (Model 4) increased specificity (78.9% vs 76.3%) but slightly decreased sensitivity (71.2% vs 72.6%). This suggests adding diagnosis codes to feature sets will decrease false positives, but increase false negatives.

### Additional testing—10,000 non-misusers

High model specificity is important for clinical implementation. The performance of Models 1 and 3 was promising towards this end. However, our small cohort (n = 930) does not represent the large population processed by universal screeners. This motivated additional testing. While the opioid misuser cohort was exhausted during training, our dataset contained additional non-misusers. From this, we decided to test each model on a large sample of non-misusers, and count the number of false positives.

We randomly selected 10,000 non-misusers and collected their diagnosis and prescription history. The data was structured similarly to the matrices for the LASSO test. Data was fed to each model and we recorded the results. Model 1 misclassified 186 of the 10,000 non-misusers (accuracy: 98.1%). Model 2 misclassified 408 non-misusers (accuracy: 95.9%). Finally, Model 3 misclassified 213 non-misusers (accuracy: 97.9%). Comparatively, these scores agree with the results in preceding sections. Model 1 demonstrated the best accuracy, Model 3 performed slightly worse, and Model 2 demonstrated the lowest accuracy. From this, we still postulate Model 1 is best for clinical implementation. Regardless, all models demonstrated strong specificity in classifying large populations. The OMPS system seems to adequately address the class imbalance problem in OUD prediction.

## Conclusions

We detail conclusions regarding our work, and make note of limitations and future directions. The results of the ten-fold cross-validation are displayed in Table 6. From these results, Model 1 performed the best (75.5% accuracy), Model 3 performed similarly (74.5% accuracy), and Model 2 performed the worst (64.9% accuracy). All models demonstrated strong specificity in large populations, suggesting OMPS enables efficient universal screening.

We believe our study is the first to focus on the problem of OUD prediction in general patient populations. Previous research primarily considered only prescription populations, as their motivation was to assist medical professionals in the prescription process. Our motivation is to aid medical practices in identifying opioid misusers before their condition worsens, which extends the misuser population to those not given a prescription. Our work suggests that the earlier identification of opioid misusers is possible. Additionally, because our models only utilize data accessible to an individual medical practice, our system can be implemented without the sharing of data between institutions. The high specificity demonstrated by our models also indicates efficient use of hospital resources. It follows that perhaps medical practices with sufficient EHR data have the capacity to identify/predict opioid misuse in their patient population.

### Limitations

We make note of limitations present in our study. First, the majority of patients in the misuser and non-misuser cohorts (94%, 92% respectively) were prescribed opioids. Our dataset had few nonprescription opioid misusers with the necessary data for our experiment. If features from non-prescription misusers differ significantly from prescription misusers, the

**Table 6. BPNN model results.**

| Model | Accuracy | Sensitivity | Specificity |
|---|---|---|---|
| Model 1 | 75.5% | 70.7% | 80.2% |
| Model 2 | 64.9% | 57.7% | 72.0% |
| Model 3 | 74.5% | 72.6% | 76.3% |

significance of our results could be impacted. Additionally, a notable portion of our opioid misusers had OUD diagnoses/prescrpitions in their history that preceeded their first misuse diagnosis. For these patients, our predictive model has less utility, as their OUD condition has already developed.

The performance of our feature sets may not generalize to other hospital settings. Previous research has illustrated significant heterogeneity between different EHR datasets [35]. Hence, ICD-CM-9 may not be common in other hospital settings. Converting our feature set to version 10 or 11 has significant challenges due to the differences in encoding detail. Additionally, medical professionals from other hospital settings may prescribe and assign codes at different rates. A diagnosis/prescription common in one setting may be rare in another. Developing unique models for specific healthcare settings may optimize performance. To our knowledge however, this has yet to be explored.

Factors unique to our experiment could have influenced results. Our opioid misuse cohort was small (465) for machine learning. This was due to restrictions imposed by diagnosis codes. For predictive modeling, choosing a single coding scheme is recommended. This requirement reduces the population and patient characteristics available for training. In our experiment, many opioid misusers were excluded from training, as they lacked ICD-9-CM codes. To be screened with Model 1, future patients must also meet this requirement. We anticipate these problems will persist with any EHR dataset containing multiple encoding schemes. Additionally, previous research demonstrates diagnosis codes are often inaccurate/inconsistent. Our results suggest encoding was consistent, but we cannot be certain. Additionally, we used a very-well managed EHR dataset (MIMIC-IV). Typical hospital settings may lack this level of data quality.

## Future directions

The following methodological changes could improve performance. First, additional markers besides non-dependent opioid misuse should be explored. We encountered many patients diagnosed with opioid dependence before non-dependent opioid misuse. Dependence is a severe OUD condition. Hence, our models do not identify these patients before their condition worsens. Constructing a separate model to predict dependence could address this issue. Second, additional techniques could be employed to increase our cohort size. Currently, our system uses only ICD codes to identify opioid misusers. It is possible that a patient was not assigned an ICD code for opioid misuse, but was identified as an opioid misuser within the doctor's notes. Therefore, employing natural language processing could increase our cohort size, providing additional data for model training.

Diagnosis codes have several inherent limitations. From this, we advocate for continued research with prescription models. Perhaps pre-processing techniques could improve performance. For example, aggregating medications into drug categories (anticonvulsants, antipsychotics, withdrawal treatment, opioids, etc.) would reduce noise. This also may increase the size of training cohorts. In our experiment patients with Nicotine Patches were included in training, but not necessarily patients with Fentanyl patches. Aggregating would include both of these patients. Finally, the number of times a drug appeared in a patient's history (drug frequency) could be explored as a predictor. Model 2 was trained to distinguish opioid misusers from non-misusers by the existence of drug combinations. Hence, patients with one Lorazepam prescription were represented identically to patients with three Lorazepam prescriptions. Considering drug frequency in training may improve performance. However, learning complexity will necessarily be increased. Experiments with large opioid misuser cohorts could explore this option.

## Author Contributions

**Conceptualization:** Lucas Vega, Winslow Conneen, Michael A. Veronin.

**Data curation:** Winslow Conneen, Robert P. Schumaker.

**Formal analysis:** Lucas Vega, Robert P. Schumaker.

**Investigation:** Lucas Vega, Winslow Conneen, Michael A. Veronin.

**Methodology:** Lucas Vega, Michael A. Veronin.

**Writing – original draft:** Lucas Vega.

**Writing – review & editing:** Lucas Vega, Robert P. Schumaker.

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
