## [Decision Letter · Decision Letter 0]

19 Mar 2024

PONE-D-24-07080A Neural Network Approach to Predict Opioid Misuse among Previously Hospitalized Patients using Electronic Health Records.PLOS ONE

Dear Dr. Vega,

Thank you for submitting your manuscript to PLOS ONE. After careful consideration, we feel that it has merit but does not fully meet PLOS ONE’s publication criteria as it currently stands. Therefore, we invite you to submit a revised version of the manuscript that addresses the points raised during the review process.

We look forward to receiving your revised manuscript.

Kind regards,

Sunil Shrestha

Academic Editor

PLOS ONE

Journal Requirements:

5. Please ensure that you include a title page within your main document. You should list all authors and all affiliations as per our author instructions and clearly indicate the corresponding author.

Additional Editor Comments:

I have thoroughly reviewed the feedback from both reviewers regarding your paper titled "A Neural Network Approach to Predict Opioid Misuse among Previously Hospitalized Patients using Electronic Health Records." Both reviewers commend the paper for addressing a significant issue, opioid misuse, and for its unique approach utilizing electronic health records (EHR) and neural network models. However, they have provided constructive feedback aimed at enhancing the clarity, coherence, and adherence to journal guidelines.

Reviewer 1 acknowledges the high value of your paper in addressing opioid misuse but notes deviations from the journal's author guidelines, leading to confusion and readability issues. They specifically highlight the lack of introduction to neural networks, EHR, and machine learning, along with unnecessary sections and inconsistent referencing styles. Additionally, they recommend integrating sections such as literature review and research questions into the introduction and methodology for better flow.

Reviewer 2 emphasizes the need for improvement in language and grammar, suggesting trimming unnecessary sections to enhance readability. They also point out specific instances of incomplete or unclear sentences, inconsistent terminology, and the absence of references for certain claims.

Both reviewers recognize the significance of your work but highlight areas for improvement in clarity, coherence, adherence to journal guidelines, and language quality. Therefore, I recommend revising the manuscript to address these concerns before resubmitting it for further consideration. Please ensure that the revisions align with the journal's formatting requirements and guidelines to facilitate the review process.

Reviewers' comments:

Reviewer's Responses to Questions

**Comments to the Author**

1. Is the manuscript technically sound, and do the data support the conclusions?

Reviewer #1: Partly

Reviewer #2: Partly

2. Has the statistical analysis been performed appropriately and rigorously? 

Reviewer #1: Yes

Reviewer #2: N/A

3. Have the authors made all data underlying the findings in their manuscript fully available?

Reviewer #1: Yes

Reviewer #2: Yes

4. Is the manuscript presented in an intelligible fashion and written in standard English?

Reviewer #1: No

Reviewer #2: Yes

5. Review Comments to the Author

Reviewer #1: Dear Authors,

After going through your paper “A Neural Network Approach to Predict Opioid Misuse among Previously Hospitalized Patients using Electronic Health Records” I can see that it has high value and address a major issue i.e., opioid misuse. The paper is unique and presents how electronic health records (EHR) could be used to predict opioid misuse using neural network approach. I agree that opioid misuse is a significant problem and can lead to cases of opioid overdose and deaths. Therefore, if earlier identification of opioid misuse could be done among previously hospitalized patients it could further benefit them from getting risk of opioid overdose.

After going through your paper, I found most of the contents within the paper is well explained and I genuinely commend authors for that. However, I also found that the authors didn’t follow the author guidelines (https://journals.plos.org/plosone/s/submission-guidelines) properly before submitting the paper. Because of the content’s irregularity, the paper can get confusing and difficult to read. I recommend authors to strictly follow such guidelines so that it maintains clarity and readability among researcher and even readers. As of current, it looks like a thesis dissertation file. Similarly, the authors also haven’t included significant components such as “the introduction of neural network, its approach, EHR in the introduction section.” This creates confusion and shows lack of coherence. I recommend authors to fully revise the paper following the guidelines. I have few comments for the authors. These comments are simple as it mainly focuses with the adhering journal guidelines along with clarity and readability of paper.

Comments:

Introduction

1. Line 11-12: “In the United States, nearly 10 million individuals misuse opioids annually National Center for Drug Abuse Statistics (2021).”

This line creates confusion on whether it’s an entire sentence or the later one is intext-citation. Please amend so that clarity can be maintained. Please maintain references thoroughly. The author guidelines mention on using “Vancouver” (See under style and format), however you seem to have used author-date style.

2. Line 12-13: “Opioid misuse occurs when an individual 1) takes opioids in different/larger quantities than prescribed by their physician, or 2) takes opioids without a prescription.”

Please cite this sentence too as this sentence provide useful information.

3. Line 35-39: With respect to your topic, I don’t find any introduction on the idea of neural network, its approach, its meaning and purpose. Similarly, I don’t see any introduction to EHR and its purpose in this sense. The authors could have also provided a little context to machine learning models and their importance in predicting such misuse.

4. Line 41-45: “The rest of this paper is structured as follows. Section 2 is the Literature Review and explores opioid risk assessment tools and machine learning. Section 3 contains our Research Questions. Section 4 is the Experimental Design and we present the OMPS System. Section 5 details the Experimental Results and Discussion. Section 6 delivers our Conclusions, Limitations and Future Directions. Finally, Sections 7 and 8 contain the Abbreviations and Declaration statements respectively”.

This is not needed. We don’t mention such things in introduction section. Better omit this.

5. Section 2 and its subsection is not required. Please follow the author guidelines properly. Within this section, you have mentioned all the overall variables/checklist (taxonomy of opioid risk assessment tools) and machine models. This perfectly sets up for the upcoming section. But a specific literature review section is not needed, please amend this portion within your introduction, methodology sections. Including this in your earlier and later sections, make your paper must robust and readable.

6. Section 2.2 Line 132-135 explains on the machine learning and Line 137-139 shows the meaning of EHR and its use. You can include this in your introduction section.

7. Section 2.3 “Research gaps” ideas could have been incorporated within the introduction section.

8. Section 3 “Research questions” could also have been incorporated within the introduction section.

9. Section 4 should be methodology not experimental design and only the authors should highlight it as an experiment design study.

10. Section 5: Please rename it to as “Result and discussion”. I acknowledge the authors idea of presenting the results as per the research questions.

11. The limitations, further implication sections are well presented in accordance with study results and objectives.

12. Please look out for potential grammatical errors, abbreviation and spelling mistakes.

I commend authors for providing such detailed explanation on the topic. However, your work needs further revision to improve its clarity and readability.

Reviewer #2: 1)Overall, the writing needs to be improved in terms of language and grammar.

2)I feel many unnecessary sections could be trimmed and included in the introduction and methodology. The manuscript is very lengthy and should be reduced to a readable pace.

3) Also, please check on the referencing system.

4)Line 11- The line seems to be incomplete. Please make sure it is clear.

5)Line 14- Opiate use disorder- Probably you could provide a clear description on this and what are the consequences.

6) Line 32- What is the definition of universal screening that you meant? Or is it based on a standardised protocol?

7) In Line 32, you mentioned that there is a lack of resources to administer questionnaires but under Figure 1 (Opiate risk assessment tools), there is questionnaire. So which one you are referring to? Please be clear.

8) Line 441- “This finding also aligns with previous research.”

Where are the references?

9) Please adhere to the manuscript formatting for the article. It looks more like a mini-thesis to me.

6. PLOS authors have the option to publish the peer review history of their article (what does this mean?). If published, this will include your full peer review and any attached files.

Reviewer #1: No

Reviewer #2: No

---

## [Author Response · Author response to Decision Letter 0]

10 Apr 2024

Response to Reviewers’ Comments

PLOS ONE

A Neural Network Approach to Predict Opioid Misuse among Previously Hospitalized Patients.

Authors: Lucas Vega, Winslow Conneen, Michael A. Veronin, Robert P. Schumaker

This document contains our response to the reviewers’ comments. Additionally, we detail revisions made to the manuscript. The reviewers both made insightful and helpful comments, especially regarding the formatting of our paper. Their guidance improved our manuscript’s readability, which we view as an invaluable improvement. As such, we are genuinely thankful for the instructive criticism provided by both reviewers. 

Additional Editor Comments:

I have thoroughly reviewed the feedback from both reviewers regarding your paper titled "A Neural Network Approach to Predict Opioid Misuse among Previously Hospitalized Patients using Electronic Health Records." Both reviewers commend the paper for addressing a significant issue, opioid misuse, and for its unique approach utilizing electronic health records (EHR) and neural network models. However, they have provided constructive feedback aimed at enhancing the clarity, coherence, and adherence to journal guidelines. Reviewer 1 acknowledges the high value of your paper in addressing opioid misuse but notes deviations from the journal's author guidelines, leading to confusion and readability issues. They specifically highlight the lack of introduction to neural networks, EHR, and machine learning, along with unnecessary sections and inconsistent referencing styles. Additionally, they recommend integrating sections such as literature review and research questions into the introduction and methodology for better flow. Reviewer 2 emphasizes the need for improvement in language and grammar, suggesting trimming unnecessary sections to enhance readability. They also point out specific instances of incomplete or unclear sentences, inconsistent terminology, and the absence of references for certain claims. Both reviewers recognize the significance of your work but highlight areas for improvement in clarity, coherence, adherence to journal guidelines, and language quality. Therefore, I recommend revising the manuscript to address these concerns before resubmitting it for further consideration. Please ensure that the revisions align with the journal's formatting requirements and guidelines to facilitate the review process.

We thank the Reviewers for noting the significance of the problem our research addresses. We also value and appreciate their suggestions regarding the structure and readability of our paper. We have made major changes in accordance with their suggestions, and believe our manuscript is better for it. Our specific revisions and reviewer responses are detailed below.

Comments - Reviewer #1

Comment 1: Line 11-12: “In the United States, nearly 10 million individuals misuse opioids annually National Center for Drug Abuse Statistics (2021).” This line creates confusion on whether it’s an entire sentence or the later one is an intext-citation. Please amend so that clarity can be maintained. Please maintain references thoroughly. The author guidelines mention on using “Vancouver” (See under style and format), however you seem to have used author-date style. 

We thank the reviewer for their observation. National Institute on Drug Abuse (2021) is meant to be an in-text citation. We have since changed our citation style to that specified by the “plos2015.bst” file, which is mandated by the journal’s latex guidelines. From this change, our manuscript is now more readable. Line 11 now reads:

In the United States, nearly 10 million individuals misuse opioids annually [1].

Comment 2: Line 12-13: “Opioid misuse occurs when an individual 1) takes opioids in different/larger quantities than prescribed by their physician, or 2) takes opioids without a prescription.” Please cite this sentence too as this sentence provide useful information. 

We agree with the reviewer and have provided a citation from the National Institute on Drug Abuse. Our manuscript now reads:

Opioid misuse occurs when an individual 1) takes opioids in different/larger quantities than prescribed by their physician, or 2) takes opioids without a prescription [2].

Where [2] references:

National Center for Drug Abuse Statistics. Opioid Epidemic: Addiction Statistics. Accessed: December 11, 2021. 2021. Available from: https://drugabusestatistics.org/opioid-epidemic/

Comment 3: Line 35-39: With respect to your topic, I don’t find any introduction on the idea of neural network, its approach, its meaning and purpose. Similarly, I don’t see any introduction to EHR and its purpose in this sense. The authors could have also provided a little context to machine learning models and their importance in predicting such misuse.

We especially thank the reviewer for this comment. We initially sought to develop the Introduction and Literature Review sections separately in order to give a full, thorough analysis of previous related work. We now realize, however, that this is unnecessary and detracts from the point of our manuscript. As this reviewer suggests in later comments, adding a few topics discussed in the literature review to the introduction would be preferable.

As such, we have removed the Literature Review section and extended our introduction to provide all the necessary context. We believe our manuscript is now more focused and digestible. 

As per the reviewer’s request, we have added a brief introduction to neural networks (lines 66-70):

A backpropagation neural network (BPNN) is a machine learning model that organizes a network of artificial neurons to learn from data in a manner that mimics the human brain [26]. In recent years, BPNN models have demonstrated superior performance over traditional ML methods in a variety of tasks. These include, but are not limited to, landslide prediction, [27] modeling quantum circuits, [28] computer vision [29] and predicting healthcare outcomes via EHR datasets [30].

We have also added an introduction to EHR (lines 51-54):

Developing a ML model using a medical practice's own Electronic Health Records (EHR) could overcome this problem. An EHR dataset is a digital representation of a patient's paper chart [26]. These datasets contain patient diagnosis history, prescription history, procedure history, admission times, and other healthcare related features.

Finally, we have added a section introducing machine learning models and their relevance to our problem (lines 39-49):

This motivates the development of faster screening methods. One potential approach involves machine learning (ML), or computer algorithms designed to learn patterns from datasets in order to produce a model. Specifically, these algorithms approximate a relationship between different feature sets and can be used to make predictions. If a ML model could be trained to predict opioid misuse with available medical records, it could automatically screen all patients within the system. Patients identified as "at-risk" by the model could then be further evaluated by a medical professional. Previous research has explored developing ML models to predict/identify various opioid use outcomes. These outcomes include prolonged opioid use (>= 90 days since first prescription) [15][16][17], an opioid overdose [18][19], opioid misuse [20] and opioid dependence (presence of withdrawal symptoms in absence/reduced amounts of opioid use) [21][22][23]24]. These studies have reported positive results, which suggests that reliable prediction of opioid use outcomes is possible. 

Comment 4: Line 41-45: “The rest of this paper is structured as follows. Section 2 is the Literature Review and explores opioid risk assessment tools and machine learning. Section 3 contains our Research Questions. Section 4 is the Experimental Design and we present the OMPS System. Section 5 details the Experimental Results and Discussion. Section 6 delivers our Conclusions, Limitations and Future Directions. Finally, Sections 7 and 8 contain the Abbreviations and Declaration statements respectively”.

This is not needed. We don’t mention such things in introduction section. Better omit this.

We thank the reviewer for their insight. We have removed this paragraph.

Comment 5: Section 2 and its subsection is not required. Please follow the author guidelines properly. Within this section, you have mentioned all the overall variables/checklist (taxonomy of opioid risk assessment tools) and machine models. This perfectly sets up for the upcoming section. But a specific literature review section is not needed, please amend this portion within your introduction, methodology sections. Including this in your earlier and later sections, make your paper must robust and readable. 

We thank the reviewer for this observation. We have revised our manuscript as suggested.

Comment 6: Section 2.2 Line 132-135 explains on the machine learning and Line 137-139 shows the meaning of EHR and its use. You can include this in your introduction section.

Revised as suggested.

Comment 7: Section 2.3 “Research gaps” ideas could have been incorporated within the introduction section.

We thank the reviewer for this comment that builds off previous suggestions to consolidate more of our paper into the Introduction. We believe our paper is more digestible/approachable from these insights. We have since removed the Research Gaps section and instead incorporated the ideas into the Introduction. 

Comment 8: Section 3 “Research questions” could also have been incorporated within the introduction section.

Revised as suggested.

Comment 9: Section 4 should be methodology not experimental design and only the authors should highlight it as an experiment design study.

We thank the reviewer for their insights and agree that "Methodology" is a more appropriate name for Section 6 than "Experimental Design" We have since changed the title of Section 4 from “Experimental Design” to “Materials and Methods” in agreement with Plos One’s formatting policies. 

The reviewer also writes "...only the authors should highlight it as an experiment design study." We are not sure what the reviewer meant by this. However, we have reworded the beginning paragraph of our “Materials and Methods” section to be clearer and more concise. We have revised the beginning of this section (now Section 3) to read:

To answer our research questions, we constructed the Opioid Misuse Prediction System (OMPS) as shown in Fig 1. In this section, we detail the major features of OMPS and report findings specific to our project. The first subsection presents the dataset used in our experiment. The remaining subsections detail the Misuser Cohort Selection, Identification of Common Features, Non-Misuser Cohort Selection, LASSO Feature Selection, Final Cohort Selection, BPNN and the Evaluation Metrics. 

Comment 10: Section 5: Please rename it to as “Result and discussion”. I acknowledge the authors idea of presenting the results as per the research questions. 

We agree that “Results and Discussion” is a fitting name for Section 5 and have renamed it accordingly. 

We thank the reviewer for their compliments regarding our discussion organization. Hopefully segmenting the discussion by the research questions makes our results clear to the reader.

Comment 11: The limitations, further implication sections are well presented in accordance with study results and objectives.

We thank the reviewer for the compliment. A major goal of ours within this research endeavor is to inspire similar work. As such, it was important to clearly communicate limitations and potential future directions.

Comment 12: Please look out for potential grammatical errors, abbreviations and spelling mistakes. 

We thank the reviewer for this observation. We have re-read the paper thoroughly and have corrected accidental grammar/abbreviation/spelling mistakes.

Comments - Reviewer #2

Comment 1: Overall, the writing needs to be improved in terms of language and grammar. 

We thank the reviewer for their observation. Numerous changes were made in accordance with this suggestion.

Comment 2: I feel many unnecessary sections could be trimmed and included in the introduction and methodology. The manuscript is very lengthy and should be reduced to a readable pace. 

This was also noticed by Reviewer #1. As detailed above, (Reviewer #1 - Comments 3, 6, 7, 8, 9) we have made significant structuring changes to improve the readability of our manuscript. We thank the reviewer for their observation.

Comment 3: Also, please check on the referencing system.

We thank the reviewer for this insight. This issue was also noticed by Reviewer #1. We have since changed the referencing style that was specified in the “plos2015.bst” file, in accordance with Plos One’s guidelines.

Comment 4: Line 11- The line seems to be incomplete. Please make sure it is clear. 

We thank the reviewer for this observation. Our previous citation format made line 11 appear to be an incomplete sentence. We have since changed our citation format to the referencing style that was specified in the “plos2015.bst” file, in accordance with Plos One’s guidelines.

We believe this fixes the clarity issue of line 11, which now reads:

In the United States, nearly 10 million individuals misuse opioids annually [1].

Comment 5: Line 14- Opiate use disorder- Probably you could provide a clear description on this and what are the consequences.

We thank the reviewer for this observation. While we did define Opioid Use Disorder (OUD) in accordance with the CDC’s definition, we failed to provide concrete examples of symptoms. This could make the concept of OUD difficult to grasp for those unfamiliar with the topic. We have since revised this section to include such examples. At lines 13-17 our manuscript now reads:

Prolonged opioid misuse often leads to an opioid use disorder (OUD) which describes the problematic pattern of opioid use that causes significant impairment or distress. Indicators of OUD within a patient include, but are not limited to, strong cravings for opioids, increased opioid tolerance, withdrawal symptoms when regular opioid consumption stops and difficulties performing tasks at work/school. Widespread OUD has resulted in an ongoing epidemic, with over 75,000 opioid overdose deaths between April 2020-April 2021 [3].

Comment 6: Line 32- What is the defination of universal screening that you meant? Or is it based on a standardised protocol?

We thank the reviewer for this observation, as we were unclear about communicating this topic. Universal screening refers to the screening of all patients indiscriminately. In the past, this terminology has been used by the Joint Comission to mandate the screening of all patients for suicide-risk. 

We have since provided an explicit description of the terminology. At lines 35-37 (different from line identified in comment due to previous revisions) our manuscript now reads:

 Any patient could potentially be an opioid misuser, but not all patients present visible signs of misuse until an overdose occurs. Thus, the thorough screening of all patients (universal screening) is necessary to address this problem.

Comment 7: In Line 32, you mentioned that there is a lack of resources to administer questionnaires but under Figure 1 (Opiate risk assessment tools), there is questionnaire. So which one you are referring to? Please be clear.

We understand now that our wording was confusing, and thank the reviewer for pointing this out. In Line 32 we wrote: However, no universal screening for opioid misuse occurs in hospitals, as they lack the resources to administer a questionnaire to every patient. We did not mean to communicate that questionnaires are not used at all in practice. They are used in screening patients that are being considered for opioid prescriptions. However, questionnaires could not be used for the screening of all patients (universal screening) because their administration would be too time consuming. This is what we meant to communicate. We included questionnaires in our 

---

## [Editor Report · Decision Letter 1]

30 Apr 2024

PONE-D-24-07080R1A Neural Network Approach to Predict Opioid Misuse among Previously Hospitalized Patients using Electronic Health Records.PLOS ONE

Dear Dr. Vega,

Thank you for submitting your manuscript to PLOS ONE. After careful consideration, we feel that it has merit but does not fully meet PLOS ONE’s publication criteria as it currently stands. Therefore, we invite you to submit a revised version of the manuscript that addresses the points raised during the review process.

We look forward to receiving your revised manuscript.

Kind regards,

Sunil Shrestha

Academic Editor

PLOS ONE

Additional Editor Comments :

Please kindly review the manuscript based on the reviewers comments

---

## [Author Response · Author response to Decision Letter 1]

1 Jun 2024

This is the fourth resubmission made to PLOS. The academic editor was unable to open our .tex file. We were unsure if this meant the file was corrupted, or if the file would not compile. We investigated the .tex files and removed obscure/unnecessary packages. We believe this, in addition to reuploading, will fix the issue. The academic editor additionally requested that we follow the latex format for our submission. We fixed this issue in our last submission, and believe the academic editor will note this once we successfully reupload our file.

We thank PLOS for their timely communication during this process. Please inform us of any other issues. Thank you.

---

## [Editor Report · Decision Letter 2]

13 Aug 2024

A Neural Network Approach to Predict Opioid Misuse among Previously Hospitalized Patients using Electronic Health Records.

PONE-D-24-07080R2

Dear Dr. Vega,

We’re pleased to inform you that your manuscript has been judged scientifically suitable for publication and will be formally accepted for publication once it meets all outstanding technical requirements.

Kind regards,

Sunil Shrestha

Academic Editor

PLOS ONE

Additional Editor Comments (optional):

ACCEPT
---

## [Editor Report · Acceptance letter]

19 Aug 2024

PONE-D-24-07080R2 

PLOS ONE

Dear Dr. Vega, 

I'm pleased to inform you that your manuscript has been deemed suitable for publication in PLOS ONE. Congratulations! Your manuscript is now being handed over to our production team.

Kind regards, 

on behalf of

Dr. Sunil Shrestha 

Academic Editor

PLOS ONE